# *Scrunch*: Preventing sensitive property inference through privacy-preserving representation learning

## Abstract

Many tasks that are commonly performed by devices attached to the Internet are currently being offloaded to the cloud, using the Machine Learning as a Service (MLaaS) paradigm. While this paradigm is motivated by the reduced capacity of mobile terminals, it also hinders privacy associated with the data exchanged over the network. Thus, the data exchanged among parties shall be conveniently anonymized to prevent possible confidentiality and privacy issues. While many privacy-enhancing algorithms have been proposed in the past, they are usually relying on very complex models that make difficult their applicability to real-world systems or envision too friendly attacker models. In this paper, we propose a deep learning system that creates anonymized representations for the data, while keeping the accuracy for the targeted MLaaS task high, assuming that the attacker can re-train an adversarial model. Our results show that the proposed algorithm i) is effective yet it uses a lighter approach than state-of-the-art ii) considers less friendly attacker models, and iii) outperforms the benchmark under different privacy metrics.

## 1 Introduction

The complexity and size of ML models is growing over time. Recent examples, such as GTP-3 with 175B parameters [OpenAI (2022 (accessed September 28, 2022)] or Megatron-Turing with 530B [Nvidia (2022 (accessed September 28, 2022)], have presented models that are impossible to generate or even maintain for most companies in the world, not to speak about academia or users with personal devices. Moreover, it is expected similar growth in the next years [Fedus et al. (2021)]. This progression, together with the slowdown in the production of new hardware severely limits the capacity of small (and even big) enterprises to use the last advances in Natural Language Processing (NLP), image recognition, or other complex ML tasks.

In this scenario, big tech companies have started to offer their models in a Machine Learning as a Service (MLaaS) fashion. That is, they run the gigantic ML models on their premises and allow customers to query the model for a pre-negotiated fare. This model is convenient for both customers that do not have the ability to create their own complex model (i.e., because they do not have a tagged dataset), and for those that need to execute (even simple) ML tasks on limited devices such as mobile phones or IoT devices.

However, to perform an MLaaS task, the customer should send the raw data (e.g., an image) to the service provider. While this operation may not present big problems in certain tasks (e.g., a connected vehicle sending telemetry data for predictive maintenance), it certainly has heavy privacy/confidentiality implications in others (e.g.., a surveillance system requesting image classification services).

Alternatively, the service provider could give the model to the customer to avoid data transfer. Nonetheless, this is typically not feasible in the case of limited devices or huge models. And even in cases where the customer could execute the model, the MLaaS provider may have concerns as the customer could blackbox or use the model without the provider's permission.

In this paper, we present *Scrunch*, a technique that allows the usage of MLaaS without the privacy implications of sending raw data to third parties. In our technique, a previously trained model is split into two parts and then its first part is fine-tuned adding a second loss function, to ensure the information after this point is only valuable to perform the task at hand, but not to perform any other task. The usage of a pre-trained model allows the easy usage of already existing models without the need of training them from scratch.

After the two parts are both trained taking into account the new, combined loss function, the first part can be sent to the customers that can execute it even with limited resources, and only transfer the obtained data representations. The rest of the model stays within the service provider ensuring that customers cannot make non-legitimate usage of the entire model from the provider.

*Scrunch* is able to create privacy-preserving data representations. It provides accuracy similar to the one of a neural network without privacy and, at the same time, provides higher privacy than state-of-the-art privacy solutions.

In the remaining of this paper, we present the privacy model in Section 2, then, we implement our solution for two different Neural Network architectures and data sets and evaluate its performance in Section 3. In Section 4, we show how the model parameters affect its way of working. Finally, Section 5 concludes the paper.

## 1.1 STATE OF THE ART

The application of privacy-preserving techniques to data sharing and ML has been widely studied in the past years with solutions ranging from the already classic k-anonymity [Sweeney (2002)], l-diversity [Machanavajjhala et al. (2007)] or t-closeness [Li et al. (2007)] to more novel solutions such as z-anonymity [Jha et al. (2020)]. Among all of them, Differential Privacy (DP) [Dwork et al. (2006)] is, with absolute certainty, the most accepted and used by the ML community.

DP grants a formal guarantee about how likely the data is to leak sensitive information, i.e. info beyond what is legitimately intended to be publicly available by the data owner. The problem to be solved in our scenario, instead, concerns 'inference privacy', i.e. reducing the amount of information that is sent/published in the first place. In addition, applying DP - e.g. in the form of noise - to the data with no further tweaks usually tends to quickly degrade the whole informational content, including what should be kept usable.

Other approaches that try to preserve the privacy of exchanged data are those that employ advanced cryptographic techniques. Two particularly researched approaches today are Fully Homomorphic Encryption (FHE) and Secure Multi-Party Computation. Thanks to FHE, direct inference on encrypted data becomes possible [Gilad-Bachrach et al. (2016); Gentry (2009); Bos et al. (2013)]. And since data is never decrypted, its privacy is guaranteed. FHE usually suffers from an accuracy drop with complex networks, since it works by approximating a neural network with a low degree polynomial function. But the real major drawback is the computational cost: the encryption schemes' complexity makes the inference time increase by many orders of magnitude, making it impractical for real-time use cases. Another cryptographic approach is Secure Multi-Party Computation (SMC), which makes it possible for two entities to compute a function over their inputs while maintaining those inputs perfectly private [Mohassel & Zhang (2017); Liu et al. (2017)]. Usually, SMC scenarios are based on garbled circuits, secret sharing, and oblivious transfer. SMC also suffers from high cryptographic complexity. Another popular field of research concerns about how to securely run an ML model on a cloud machine. Proposals from this field rely on trusted execution environments such as Intel SGX and ARM TrustZone [Tramer & Boneh (2018); Hanzlik et al. (2021)]. Nevertheless, such scenarios still require the client to trust the cloud servers with their data.

Finally, more similar to our work, there is another sub-field of the Privacy-Preserving research community that tries to generate privacy-preserving data representations. AutoGAN [Nguyen et al. (2020)] proposes a non-linear dimension reduction framework based on a GAN structure. On it, a Generator and a Discriminator are iteratively trained in turn, in an adversarial manner, to enforce a certain distance between original and potentially reconstructed data. Osia et al. (2018) protects the data against the execution of previously known ML tasks. Moreover, works like [Osia et al. (2020)], Cloak [Mireshghallah et al. (2020a)] and Shredder [Mireshghallah et al. (2020b)] apply a

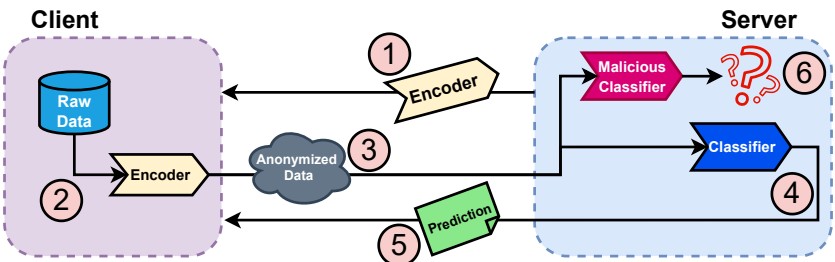

Figure 1: High-level scenario.

contrastive loss to privatize the data with small differences among them on the network structure and application. Contrary to all of them, we employ the center loss in our work, allowing that way better privacy protection without the need of complex Siamese Networks.

## 2 MODEL

The *Scrunch* model uses as a starting point a pre-trained neural network, then it splits it into two parts and makes modifications to it to improve the privacy provided by the final solution. In this solution, the computation of the inference is shared between a client, that executes the first part of the neural network (namely, the *Encoder*) and sends the output features to a Server (i.e., in the cloud) that executes the second part (namely, the *Classifier*) and returns the ML task answer.

Figure 1 shows a summary of how the general model works in inference: 1) The server sends an *Encoder* to the Client. 2) The client uses the encoder to generate anonymized representations of the raw data. 3) The client sends the anonymized data to the Server. 4) The server uses the *Classifier* to solve the ML task at hand and 5) sends the prediction back to the client. 6) Even using the data available to the Server, the latter - or another, malicious entity - cannot train a different *Classifier* to solve a different task, based solely on the privatized representations.

Following, we describe how to efficiently generate an *Encoder* and a *Classifier* to maximize the accuracy for the task to be solved, while decreasing the performance of other non-related tasks over the anonymized data.

### 2.1 ENCODER AND CLASSIFIER

We build on the intuition that features obtained in the middle of a neural network are an abstract representation of the input data and can be used to share data in a privacy-preserving way. However, without additional modifications, such features may still contain a significant amount of extra information, as demonstrated for example by the visualization techniques [Dosovitskiy & Brox (2016)].

Moreover, it is well known that, the deeper we go through the layers of a neural network, the more specialized, abstract [Yosinski et al. (2014)] and specific to the main task the features become. Furthermore, in most networks design, going deeper means also having to deal with much fewer dimensions. This all collaterally contributes to data privacy: whatever extra information was contained in the data – beyond what is actually useful to perform the main task – goes gradually and irreversibly lost [Osia et al. (2017); Malekzadeh et al. (2018)].

In the extreme case, when the network split is done after the last layer, the complete ML task would be executed by the client, obtaining that way "perfect privacy". However, this situation is not realistic in most of the use cases, either because the client cannot run the complete neural network due to hardware limitations, or because the service provider does not want to share the complete model with the client.

Thus, as a rule of thumb, one should first choose the split point that provides the client with the heaviest model it can support (and the service provider is willing to share). This alone would already grant some degree of privacy – depending on the chosen point – regarding any other extra

information the data carries. Then, in addition to this – especially in cases where one is constrained to the lower levels of the network –, other approaches may further enhance the privacy of the data.

## 2.2 ADDING PRIVACY

In a scenario where no knowledge about potential sensitive features is assumed (i.e. no private labels), the only reasonable choice is to try and reduce the general amount of information contained in the data, while constraining the accuracy of the main task to be as high as possible. In a more formal way, considering the input data and their corresponding privacy-preserving representations, we would like to reduce their Mutual Information (MI) as much as possible, while still keeping the cross-entropy loss of the whole model as low as possible. The latter is embodied in the typical softmax cross-entropy loss, computed on the output of the very last layer of the intact model: it basically keeps the points belonging to different classes well separated in the encoding space. In other words, it tries to maximize the inter-class distance.

In addition to this, we would also like to minimize the inter-class distance, to reduce any extra information contained in the structures of same-class points. Other works have tried to achieve this by employing siamese network architectures and contrastive or triplet losses [Osia et al. (2020); Mireshghallah et al. (2020b)]. The problem is these methods [Sun et al. (2014); Wen et al. (2016a); Schroff et al. (2015)] suffer from a non-negligible data expansion since the training set must be recombined in specific pairs. Furthermore, since these losses need pairs of points to be computed, two forward passes are needed, thus increasing the training time. That is why we chose instead to employ the Center Loss function, originally proposed by [Wen et al. (2016b)]. It is worth noting that, same as for triplet and contrastive losses, this loss was not primarily designed for privacy-preserving purposes, but to improve the discriminative power of the deeply learned features. As far as we know, this is the first time it is actually applied to such a goal. The center loss is defined as:

$$\mathcal{L}_C = \frac{1}{2} \sum_{i=1}^{m} ||x_i - c_{y_i}||_2^2 \tag{1}$$

where $m$ is the total number of classes in the dataset, $x_i$ is the i-th encoding, and $c_{y_i}$ is the center of the $y_i$ class. Hence, the term introduced by $\mathcal{L}_C$ minimizes the euclidean distance between any encoding classified to that class and the center of such class. We apply this function as done by Wen et al. (2016b), combining it with the usual softmax categorical cross-entropy $\mathcal{L}_S$. The two losses are governed by a weight factor $\lambda$ as follows[1]:

$$\mathcal{L} = \lambda \mathcal{L}_C + (1 - \lambda)\mathcal{L}_S \tag{2}$$

In other words, we train the model with Joint Supervision, as discussed in the work by Wen et al. (2016b). The variable $\lambda$ plays a fundamental role in steering the encoding anonymization process, and weights how the encodings are anonymized. First, we remark that these two terms need to be jointly optimized (so, configurations where $\lambda$ is 0 or 1, are asymptotic cases that are not necessarily meaningful in a real scenario), but $\lambda$ also has a meaning for the anonymization process. When $\lambda = 0$, the model is trained to only optimize the cross-entropy among classes, as a typical classifier would do. Thus, encodings may still exhibit intra-class variations that may leak sensitive information, as shown by Dosovitskiy & Brox (2016). Basically, when $\lambda = 0$, no anonymization is applied beyond what is already provided by the deep features. Instead, when $\lambda = 1$, no effort is made by the network to separate the classes, thus yielding poor results on the machine learning task. Hence, the correct parametrization of $\lambda$ is fundamental to steering the operation of the system toward an optimal balance between accuracy and privacy.

Assuming we have a pre-trained model for our 'Public Task', and once the split point has been decided, Figure 2 presents - in this case specifically for the VGG16 model - an example of how our pipeline would work when modifying the architecture:

1. The architecture is rearranged so that it becomes a multi-output network, with the activations of the split point going two separate ways:
   - The natural continuation of the original network, i.e. the Dense Part

---

[1]For more details about the loss function implementation, please refer to Wen et al. (2016b)

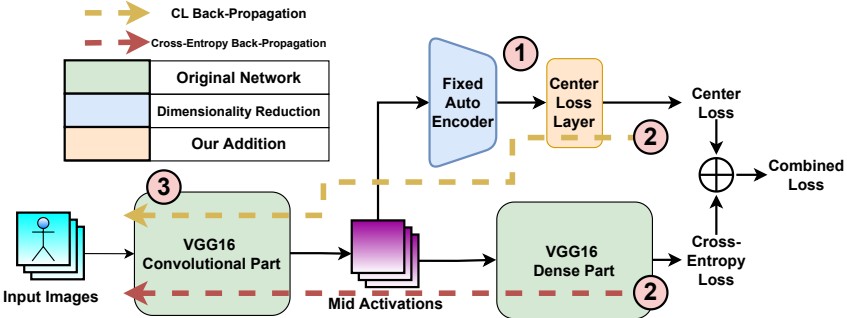

Figure 2: Model Architecture

- The new added branch, composed of a fixed pre-trained Autoencoder – for dimensionality reduction – followed by the main component of the approach, i.e. the Center Loss Layer

2. The whole network is carefully fine-tuned with a low learning rate and a Joint Supervision

  - During the forward pass, the CL layer "learns" new, more accurate class centroids, based on the current batch. Such centroids are then used to compute the Center Loss
  - The weights in the Dense part are adjusted thanks to the Cross-Entropy back-propagation, as usual
  - The Convolutional Part, instead, is trained by the back-propagation of both losses, so that it produces features that retain both privacy and utility

3. Once the training is finished, the first part of the network will be frozen and deployed on the client side as a black-box anonymizer

## 3 EXPERIMENTAL EVALUATION

We implemented *Scrunch* to work with two different network architectures and datasets. Furthermore, we tested it against some of the privacy metrics that are in line with the attacker model presented in Sec. 1, benchmarking it against the results obtained by another privacy-preserving solution: Shredder. For this, we re-implemented the original Shredder source code in Tensorflow/Keras, as also *Scrunch* is implemented using the same framework.

### 3.1 DATASETS AND NETWORK ARCHITECTURES

We employ two different architectures – both for image classification – to test our approach: a LeNet-5 neural network with the MNIST dataset, for which we also provide a direct comparison with another state-of-the-art approach [Mireshghallah et al. (2020b)] and a VGG16 neural network, with the CelebA dataset.

The LeNet-5 network takes as input a 32 X 32 X 1 image. The channel is just one because the network is designed to use greyscale images. The architecture then consists of 5 learnable layers. The first three of these are convolutional (Conv) layers with 6, 16, and 120 filters, respectively. All three layers use a kernel size of (5,5). After the 1st and the 2nd Conv layers, we also find a Max Pooling (MP) layer. Finally, the last two are Fully Connected (FC) layers. The first has 84 neurons, while the second and last FC layer has usually 10 neurons since the digits in the MNIST dataset are 10. Lastly, such a layer is usually followed by a Softmax layer that classifies the images into the corresponding classes.

The MNIST dataset [LeCun & Cortes (2010)] used to test this case is composed of greyscale images of size 32x32 pixels, representing handwritten digits going from '0' to '9'. The training set contains 60,000 different examples of images, while the test set contains another 10,000 example images for model evaluation. The labels are simple integer numbers from 0 to 9, that indicate the specific digits in the images

The other network in use is the well-known VGG16 [Simonyan & Zisserman (2014)], a 16-layer-deep CNN (Convolutional Neural Network) primarily designed to perform image classification. In particular, we use the pre-trained version of VGG16 – trained on the huge ImageNet dataset [Deng et al. (2009)] – and we first fine-tune it for the 'public task', via transfer learning. The network consists of a first, convolutional part, and then a second, fully connected part. The first part is composed of 5 macro-blocks. Each of these blocks is a stack of 2 or 3 Convolutional layers, always followed by a Max Pooling Layer. For each block, all the conv layers have the same number of filters. From the 1st to the 5th block we have, respectively, 64, 128, 256, 512, and 512 filters. The 2nd part of the network is simply made up of a Flatten layer followed by two Dense-Dropout blocks and the final Dense Layer. The first two Dense layers have both 4096 neurons, and both dropouts have a probability of 0.5, while the last Dense Layer has a number of neurons that depends on the number of classes of the specific task at hand.

In this case, we use the CelebA dataset [Liu et al. (2015)]. The original CelebA dataset consists of 202,599 images of celebrities' faces, of variable sizes. We crop these images to a fixed size of 128x128x3 pixels. The images are colored, hence the three channels. The dataset comes with a huge set of labels: 10,177 identities; 5 landmark locations for each image; 40 binary attributes annotations per image. We limit ourselves to two sets of binary labels: the gender and a label that indicates whether the person in the photo is smiling or not. We use gender as our primary/public task, and purposefully choose a simple binary attribute such as smiling/not-smiling as our malicious/private task. We do this in order to prove that our approach works for hindering even such a simple task, as opposed to choosing something that would more likely be private information – and intuitively more difficult to leak – such as identity.

## 3.2 PRIVACY METRICS

**Mutual information (MI)** is an information-theoretic notion. Assuming two data points x and y, MI(x,y) quantifies the average amount of information that leaks (i.e. is learnable) from y about x. This measure is commonly employed in literature [Cuff & Yu (2016); Kalantari et al. (2017)], both as an anonymity metric when dealing with database potential leakage [Wang et al. (2016); Liao et al. (2017)] and to better explain the behavior of neural networks [Saxe et al. (2019)]. In our experiments, we calculate the Mutual Information Drop experienced when comparing the MI between raw data and simple mid-network features (no enhanced privacy) with the MI between raw data and the data representations obtained after the encoder.

Moreover, since the encoder is public (to all possible clients, and the server itself), a malicious entity could try to retrain a Classifier in order to solve a different task than the one requested by the Client. Thus, we compute the normalized **Private Task Accuracy Drop** of an adversary classifier. That is, how worse the privacy task results are when trained on deep features or data representations, with respect to a typical classifier free to train the private task on the raw input data.

Proving that the chosen 'sensitive' task fails does not guarantee that any other potentially sensitive information is safe, of course. Hence, we resort to also testing how well a full decoder can be trained on some leaked data, in order to reconstruct the original input images from the data representations. Intuitively, if the reconstructed images are too similar to the original ones, it means that the extra information contained in the image may potentially leak from the data representations obtained. We not only provide a visual comparison for such similarity but also a quantitative measure, in the form of the **Structural Similarity Index Measure (SSIM)** [Wang et al. (2004)], a metric specifically designed to mimic human perception of 'image similarity'. In its simplest form, SSIM is represented by a number between -1 and 1 - usually rescaled between 0 and 1 - where -1 means "completely different" and 1 means "exactly the same".

The architectures of the adversary classifier and decoder are the following:

- For the classifier, we take what remains of the VGG16 or the LeNet-5 network after the chosen split point, and retrain it with the private labels and the anonymized data representations of a holdout set.

- For the decoder, we employ a simple custom upsampling network for the LeNet-5/MNIST case and the AlexNet decoder [Dosovitskiy & Brox (2016)] for the VGG16/CelebA case.

Table 1: Accuracy and privacy results.

| Dataset | Algorithm | Normalized Public Task Accuracy (%) | Private Task Accuracy Drop (%) | Mutual Info Drop (%) | 95th Percentile SSIM Drop (%) |
|---------|-----------|-------------------------------------|--------------------------------|----------------------|-------------------------------|
| MNIST | NoPrivacy/DeepFeatures | 100.0 | 00.36 | N/A | 10.19 |
| | Shredder | 98.93 | 23.98 ($1.17^2$) | 66.97 | 91.34 ($75.57^2$) |
| | *Scrunch* | **99.34** | **60.56** | **79.66** | **81.75** |
| CelebA | NoPrivacy/DeepFeatures | 100.0 | 29.86 | N/A | 39.94 |
| | *Scrunch* | **99.73** | **40.02** | **29.84** | **70.8** |

We train it with a holdout set of anonymized data representations, and their original counterparts as labels.

## 3.3 PRIVACY RESULTS

Table 1 shows the obtained privacy results against the metrics we introduced above. Higher values mean better performances, in all columns. The public accuracy is normalized by the accuracy that the models reach in absence of any privacy-preserving approach, for their public tasks. The private accuracy loss is computed with respect to the accuracy that a full, free model would obtain by training with the private labels. We tested two popular datasets, MNIST and CelebA, using LeNet-5 and VGG16 respectively as the backbone network for the classification tasks. Both networks have been split right after their convolutional part (layer block5_pool for VGG16 and conv_3 for LeNet-5). For the training of the encoder, we split the original dataset into Training, Validation, and Test with a 40%, 40%, 20% proportion for both the solutions. When training the adversary attacker classifiers we also use a similar split. For *Scrunch* the best results are obtained with $\lambda = 0.9$, with a learning rate of $10^{-5}$. The adversary classifiers and decoders are trained with learning rates of $10^{-5}$ and $10^{-3}$, respectively. Shredder is configured as in the original paper.

Results show that for both the datasets, *Scrunch* can basically retain all the accuracy on the public task as if there was no privacy solution applied, with a drop that is below 1% for MNIST and 0.26% for CelebA. Also, Shredder is on par with these results. However, *Scrunch* proves to be much more powerful in dropping the accuracy in the private tasks than Shredder. For the MNIST case, where both implementations can be compared, we more than double the privacy drop obtained by Shredder, approximating the performance of a random guesser.

This is also partially explained by the loss in the mutual information (13% higher in *Scrunch*), which quantifies the amount of information - about the original data points - that leaks from the data representations. Similar considerations apply also to the CelebA dataset (for which we do not have a working Shredder implementation). Here we can drop the accuracy on the private task, while the mutual information is less affected due to the high dimensionality of the input raw data.

It is fundamental to remark here that, besides obtaining better privacy metrics than a state-of-the-art solution, *Scrunch* does that while enforcing a much more stringent privacy scenario. While Shredder does not retrain the adversarial attacker on possibly similar data with available labels (an aspect that is more appropriate in many cyber-attack scenarios), in our approach we let the adversary re-train on leaked data, to improve the effectiveness of the attack. Still, even in this extremely challenging scenario, we can obtain better results than Shredder for MNIST and a significant decrease in the accuracy for the private task for CelebA.

Figure 3 provides an overview of the visual reconstructions. Without privacy, the raw data can be reconstructed in a more or less straightforward way from the decoders, depending on the network complexity and the chosen split point. Particular mention should go to the VGG16/CelebA case (right image): while the simple embeddings - i.e. with no privacy - are unmistakably much less recognizable than their original counterparts, it is still possible, even for a human observer, to differentiate sensitive features (e.g. whether the person is smiling). Such a sensitive feature is then lost when Scrunch is applied, leaving the public task the only one feasible. The left image provides a quick intuition of how *Scrunch* works with respect to Shredder and other naive solutions. *Scrunch* "learns" how to blend together samples from the same public classes, making them indistinguishable

---

[2]This is the obtained value when the adversary classifier is retrained with a holdout pool of data

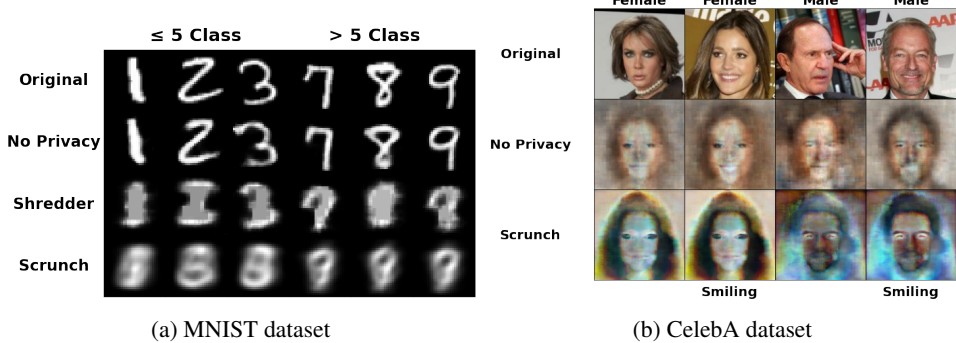

(a) MNIST dataset          (b) CelebA dataset

Figure 3: Example images reconstructed from the data representations obtained after the *encoder*.

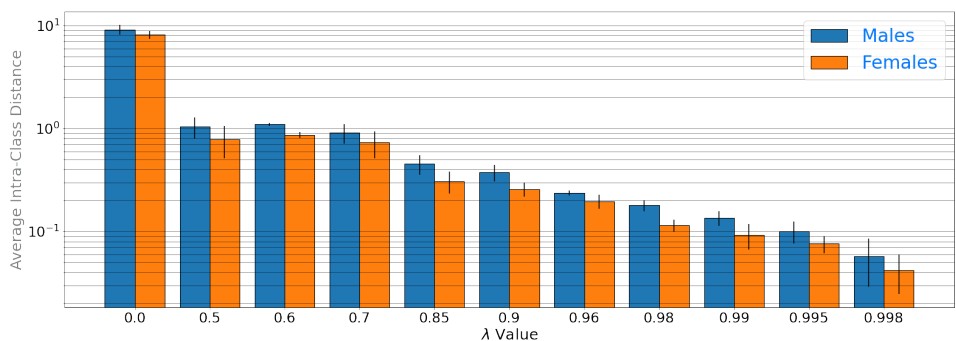

Figure 4: Intra-Class distance for different values of $\lambda$

from the private ones. Shredder, instead, works by learning "where to add noise" on each sample, a technique that proved to be less resilient in our benchmarks.

## 4 PARAMETRIZATION

As discussed in Section 2, *Scrunch* can be steered through the configuration of $\lambda$. While in Section 3 we discussed the results with the best $\lambda$, in this section we discuss the effect of $\lambda$ to steer the operation of the system, using the datasets introduced above.

In *Scrunch*, $\lambda$ is used to weight the categorical cross entropy and the center loss. Figure 4 depicts the intra-cluster distance for the two public categories of the CelebA dataset. If we take as reference the results for $\lambda = 0$, increasing it allows for reducing the intra-cluster distance by order of magnitudes. By reducing the intra-cluster distance, using the same data for tasks other than the public one results in a lowered accuracy, as discussed earlier.

We further dig into the result by analyzing how $\lambda$ affects the utility versus privacy trade-off, that is how the data transformation for privacy purposes affects the accuracy of the public task. We depict it in Figure 5, indicating the confidence interval in the accuracy averaged over 10 repetitions for the CelebA dataset. We can clearly observe two areas: the first for higher $\lambda$ that yields higher privacy at a cost of lower public accuracy. The public accuracy plateaus instead in the second region, where decreasing $\lambda$ does not help the public accuracy, while it quickly increases the private one. Thus, by changing $\lambda$ we can effectively control this trade-off.

Another way of assessing how the inter-class distance is reduced by lambda is by plotting the 2-D embeddings using t-SNE for different $\lambda$ values. We depict it in Figure 6 for the MNIST dataset, where the upper row is colored to represent the clusters for the private task and the bottom row is representing the public one. While there is a clear separation for lower $\lambda$ for both public and private clusters, the effect of a growing $\lambda$ blends points belonging to the same public cluster but different private ones, achieving thus a drastic drop in accuracy for the private tasks shown in Table 1.

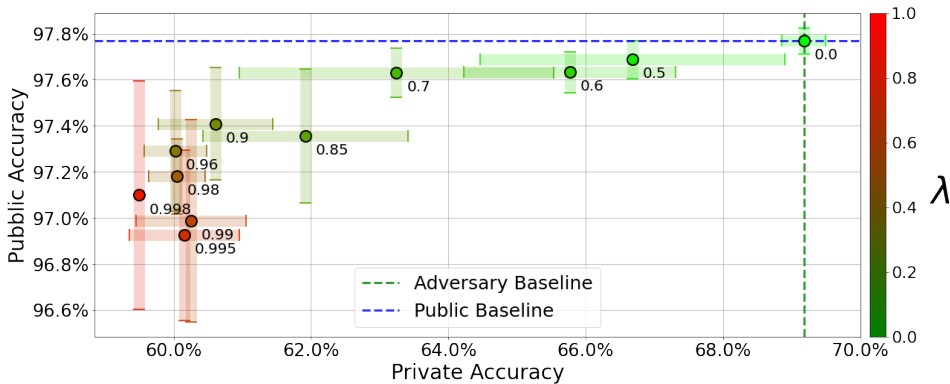

Figure 5: Utility/Privacy Trade-off

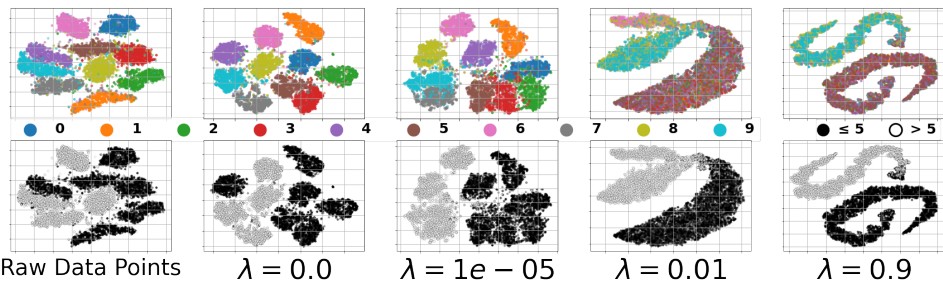

Figure 6: t-SNE of encodings for different Lambda values (MNIST dataset)

## 4.1 LIMITATIONS

While an increase in the value of $\lambda$ comes with a significant increase in the privacy obtained, it also comes with a major difficulty for the neural network to converge during the training. For example, over 70% of the training phases failed to converge - and started overfitting, triggering an early stop - after 70 epochs (on average) when $\lambda = 0.999$, while all the trainings converged in less than 40 epochs for $\lambda = 0.7$. Moreover, the average time to complete training also grows with $\lambda$. Still, while the increase in training time appears to be linear, the decrease in the intra-cluster distance is exponential.

## 5 DISCUSSION AND CONCLUSION

This paper presents *Scrunch*, an ML technique to learn privacy preserving data representations of data that can be used in MLaaS scenarios. In *Scrunch*, the client locally executes an *encoder* over the raw data and only shares the privatized data representations with the MLaaS Server, ensuring that way that the server (or other malicious entities) cannot use the data to perform any task different than the one requested by the Client.

In *Scrunch*, the *encoder* used by the client, and the *classifier* used by the server are generated by splitting an already trained neural network, modifying it by adding a new layer that ensures the intra-class distance minimization (i.e., the center loss) and retraining the whole model using joint-supervision.

*Scrunch* has demonstrated to provide much better privacy than state-of-the-art solutions (38% and 13% in Private Task Accuracy Drop and Mutual Information Drop, respectively) while keeping the accuracy for the public task virtually unaffected in spite of ensuring a much more realistic privacy model. Finally, we show how *Scrunch* can be parameterized to steer its operation, for instance by trading privacy for accuracy and reducing the training time.

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
