# OpenReview forum: "Scrunch: Preventing sensitive property inference through privacy-preserving representation learning"
_ICLR.cc/2023/Conference — Submitted to ICLR 2023_

### Official Review · Reviewer_rRGt · 2022-10-21

**Confidence:** 2
**Correctness:** 2
**Technical Novelty And Significance:** 2
**Empirical Novelty And Significance:** 2
**Recommendation:** 3

**Clarity, Quality, Novelty And Reproducibility:**

Clarity&Quality: Be honest, I dont quite understand why the model can guarantee privacy.

Novelty: The idea to use features in the middle of neural networks is rare, but in some extents, features themselves are not enough to guarantee data privacy.

**Strength And Weaknesses:**

Strength:

1. Accuracy performance is not significantly affected.

Weakness:

1. This work does not define security model.

2. Using middle features of neural network is still risky to leak information. For example, if client data are collected from smart watch, which contains physiological time-series data, after encoding, it may still contain trends how data change in time.

**Summary Of The Paper:**

As MLaaS rises on cloud, data privacy becomes an issue. Clients dont want to send their raw data directly to the server and server does not want clients to use ML models without permission. This work proposes a framework named Scrunch by using encoder sent from server for clients to generate anonymized representations of raw data and preventing malicious client to train another one for other tasks.

**Summary Of The Review:**

The idea to use features generated by neural network is a good perspective, but this work is considering less on security model and features themselves are still risky for information leakage. Except the smart watch example, I think the proposed model does not even work for NLP applications. As we know, words like “the” will appear frequently and encoding vectors will be quite similar if server provides the same encoder. It is possible for server to deduce frequent words from received features and server is also able to “reverse engineering” whole sentences or passages with cryptographic analysis. Upon the generated features, I think they still need to apply somehow encrypted techniques before sending. Hence, I think this work is not sufficient to address data privacy issue for MLaaS systems on the cloud.

---

### Official Review · Reviewer_QZoH · 2022-10-28

**Confidence:** 5
**Correctness:** 2
**Technical Novelty And Significance:** 2
**Empirical Novelty And Significance:** 2
**Recommendation:** 3

**Clarity, Quality, Novelty And Reproducibility:**

Clarity: The paper is generally clear and easy to read and follow.

Quality: The quality of the work is also poor and the reasons are discussed under Weaknesses.

Novelty: The novelty of the work is somewhat limited from a foundational point of view. The core idea is similar to split learning, which surprisingly has not been mentioned in this paper.

Reproducibility: The paper seems to be fairly reproducible, as the authors mention the various hyperparameter settings used. However, details such as optimizers used and number of training epochs are missing.

**Strength And Weaknesses:**

Strengths:
1) The paper is clear and mostly easy to understand
2) The problem statement is well-defined
3) Good use of figures and tables
4) Interesting evaluation metrics

Weaknesses:

1) The proposed approach is severely flawed due to the following reasons.

a) Firstly, the title and abstract suggest that the goal is to perform privacy-preserving "inference" in the MLaaS scenario. However, the proposed approach requires finetuning the encoder component, which can done only during the "training" phase.

b) Secondly, to perform this training (to compute the cross-entropy loss), the server needs access to the private labels of the client along with the "smashed" representations. Next, to compute the center loss, the server also needs access to the class centers. Finally, the encoder is initially shared by the server to the client and the server also knows the loss at the split point for each iteration. Using this information, the server can easily recreate a "surrogate" encoder, which will be exactly identical to the client encoder. Thus, the server has all the information including the entire neural network, class labels, and class centers in the feature space. The only unknown is the input samples, which should be easy to reconstruct using well-known "model inversion" attacks.

c) Finally, and in fact fatally, the threat model for the proposed approach is completely messed up. If the assumption is that the server is "curious" to know about the raw data, why would it implement the center loss honestly? The server can very well turn off the center loss to zero and back propagate only the cross-entropy loss and the client has no way of verifying it. Moreover, the encoder becomes privacy-preserving only for the inference phase. Thus, there will no defense against privacy leakage during the training phase.

2) Following are some of the other issues with the paper:

a) There are several ambiguous statements, e.g. “In other words, it tries to maximize the inter-class distance. In addition to this, we would also like to minimize the inter-class distance, to reduce any extra information…” should probably have said intra-class distance in the latter part instead. These errors affect the clarity of the paper and may negatively affect appreciation of its contributions.

b) What is the private task for the MNIST dataset? It is well-stated for CelebA, together with supporting visualizations, but this information is not provided for MNIST although results for the private task accuracy drop on MNIST are reported in Table 1.

c) The experimental evaluation seems somewhat simplistic: the paper claims that the proposed method is effective at preventing privacy invasion via expression recognition (smiling vs not smiling) from a public task of gender recognition (which is also binary and arguably naturally privacy-preserving). However, proof in a more challenging setting (from both public and private task perspectives) e.g., identity inference (public) vs gender (private) would be more convincing, as such challenging settings are likely to be of more real-world interest to attackers.

d) The two models evaluated in this work are relatively shallow and architecturally-similar (basic CNNs). It would be more beneficial to consider more sophisticated architectures such as ResNets, DenseNets or Vision Transformer, which are significantly deeper and more commonly-used architectures today. Some experimentation extending the proposed method to actual large-scale networks such as those mentioned above would provide compelling evidence about the benefits of the method.

e) The paper notes that the choice of lambda is particularly important. More investigation on this point would be a good addition.

**Summary Of The Paper:**

The paper presents a new approach called "Scrunch" to perform privacy-preserving training in MLaaS settings. The key idea is to split the model architecture into two parts (one for client called Encoder and another for the server called Classifier) and optimize both parts to maintain predictive performance while explicitly minimizing information leakage via Center Loss. The paper presents results on two models (LeNet-5 and VGG-16) and two datasets (MNIST and CelebA).

**Summary Of The Review:**

The paper is well-written and easy to follow, but has serious technical flaws and the results are unconvincing.

---

### Official Review · Reviewer_2RqB · 2022-10-31

**Confidence:** 4
**Correctness:** 3
**Technical Novelty And Significance:** 2
**Empirical Novelty And Significance:** 1
**Recommendation:** 3

**Clarity, Quality, Novelty And Reproducibility:**

The paper is reasonable but there are some grammatical errors (perhaps through out the paper).. For instance,

“last advances” ==> “latest advances” page 1, intro, para 1
"It is fundamental to remark" .. you probably mean "it is important to remark"..

The novelty is quite incremental..

**Strength And Weaknesses:**

I do not think the idea of splitting a network into two parts is a new one, especially in the context of privacy. The idea of adding additional constraints in the encoding process to control mutual information flow is an interesting take. Unfortunately, I am not convinced about its effectiveness.

For a primarily emprical paper, the work considers rather simplistic datasets and only two of them. In the second dataset, they do not compare it with any baselines and in the first dataset it is compared with 1 baseline. There are lot of other experiments that can be performed and baselines that can be and should be compared with. For example, why not compare with fully homomorphic encryption based approaches?

Overall I think the paper is rather incremental with no “aha” factor that is needed in a conference such as ICLR.

**Summary Of The Paper:**

This work studies the problem of privacy preserving representation learning. The goal here is to enable secure inference without revealing the client data. The central idea is to split a neural network into two parts — and sending the first part to client. The client applies raw data through the first part ofthe neural network (with some additional loss function) and then sends the encoded data to the server for final inference. The key idea is that by controlling the information at the encoding stage privacy of protected attributes is preserved.

The experimental analysis compares this method with another method in prior art, namely Shredder, and the non-private counterpart. There two metrics of privacy employed — one to see how much information is revealed by the first part and second one to see the see how much information is revealed in the encoding step.

**Summary Of The Review:**

I think the paper is an incremental effort over the state of the art methods. The experimental section is rather limited for an experimental paper. It needs a lot more benchmarks and results on multiple datasets and baselines.

---

### Decision · Program_Chairs · 2023-01-20

**Decision:**

Reject

**Justification For Why Not Higher Score:**

The method is flawed, the experiments are not sufficient, and the writing can be improved.

**Justification For Why Not Lower Score:**

N/A

**Metareview: Summary, Strengths And Weaknesses:**

In this paper, the authors develop a method called "Scrunch" for MLaaS privacy-preserving. The key idea is to split the neural network model into two parts, the first part called Encoder is sent to client, and the second part called Classifier is for the server. Both parts are optimized for good accuracy while minimizing information leakage.  The task and the proposed approach appear interesting. However, reviewers consider the method is flawed, the experiments are not convincing, and the writing can be improved.  The authors are suggested to improve the paper by incorporating reviewer's constructive comments.